# Nb-Based Catalysts for the Valorization of Furfural into Valuable Product through in One-Pot Reaction

**DOI:** 10.3390/ijms25052620

**Published:** 2024-02-23

**Authors:** Rocío Maderuelo-Solera, Benjamín Torres-Olea, Carmen Pilar Jiménez-Gómez, Ramón Moreno-Tost, Cristina García-Sancho, Josefa Mérida-Robles, Juan Antonio Cecilia, Pedro Maireles-Torres

**Affiliations:** Department of Inorganic Chemistry, Crystallography and Mineralogy, Málaga University, 29071 Málaga, Spain; rocioms@uma.es (R.M.-S.); benjamin@uma.es (B.T.-O.); carmenpjg@uma.es (C.P.J.-G.); rmtost@uma.es (R.M.-T.); jmerida@uma.es (J.M.-R.); maireles@uma.es (P.M.-T.)

**Keywords:** furfural, niobium catalysts, furfuryl alcohol, alkyl furfuryl ethers, one-pot processes, hydrogen catalytic transfer

## Abstract

Nb-based catalysts supported on porous silica with different textural properties have been synthesized, characterized, and tested in the one-pot reaction of furfural to obtain valuable chemicals. The catalytic results reveal that the presence of fluoride in the synthesis, which limits the growing of the porous silica, limits diffusional problems of the porous silica, obtaining higher conversion values at shorter reaction times. On the other hand, the incorporation of NbO_x_ species in the porous silica provides Lewis acid sites and a small proportion of Brönsted acid sites, in such a way that the main products are alkyl furfuryl ethers, which can be used as fuel additives.

## 1. Introduction

Nowadays, the depletion of fossil fuels has prompted the search and development of new and more sustainable processes of energy and chemical production. Amongst them, the scientific community and governments have paid attention to the use of biomass, since it is the only source from which they can be obtained, becoming an alternative to traditional fossil-based resources. However, the selection of biomass must be carried out carefully, avoiding to interfere with the food chain, which would cause speculation and increasing social imbalances.

In this century, lignocellulosic biomass has emerged as a sustainable resource because, in many cases, this comes from agricultural residues with low commercial value [1]. Although the composition of lignocellulose depends on the species, even on the growing and harvesting conditions, it is generally composed of cellulose (40–50%), hemicellulose (25–30%) and lignin (15–20%) [2]. Different physical and chemical processes have been proposed for the fractionation of lignocellulose in their components to facilitate its subsequent valorization [3]. Focusing on hemicellulose, in general, this is a biopolymer formed mostly by C5 sugars, mainly xylose and arabinose, linked through β-(1→4)-glycosidic bonds. This biopolymer can be hydrolyzed in their respective monomers by hydrothermal acid treatment [4]. The C5 monomers can be subsequently dehydrated under acid conditions to produce furfural (FUR), one of the most important platform molecule derived from biomass [5].

FUR is considered as a versatile molecule of great potential, because its chemical structure confers a high reactivity, being an intermediate for a wide variety of high value-added chemicals [6]. In this sense, FUR hydrogenation to furfuryl alcohol (FOL) accounts for 62% of the use of FUR produced, because FOL is an excellent monomer for the synthesis of resins with high thermochemical stability, as well as for the synthesis of other valuable chemicals [7]. However, the interest in these hydrogenation reactions is not limited to the synthesis of FOL, since it is possible to obtain other chemicals with potential applications as fuel additives, solvents, monomers, or starting products for fine chemistry production [7]. Traditionally, the hydrogenation of FUR has been carried out using transition metals as the active phase. The catalytic activity and selectivity pattern depend on the hydrogenation capacity of the metal used as the active phase, as well as the acid-base properties of the support in the case of supported metal catalysts [8]. Among the catalysts used in these reactions, the relevance of Cu-based catalysts must be highlighted due to their low cost, high activity and selectivity towards furfuryl alcohol or 2-methylfuran [7,8,9,10], but other metallic phases, such as Ni, Co, Pd, Pt or Ru, have also been reported in the literature [6,8,11,12,13,14,15], leading to other valuable chemicals, such as tetrahydrofurfuryl alcohol, tetrahydrofuran, furan, hydrocarbons or alcohols [6,7,8]. In recent years, FUR reduction processes have also been performed through catalytic transfer hydrogenation (CTH) using a broad spectrum of catalysts [16]. These CTH processes involve the reaction between an aldehyde or ketone and a sacrificing alcohol, promoted by the presence of Lewis acid sites, where the hydrogen is transferred through a six-membered intermediate ring, according to the Merweein-Ponndorf-Verley mechanism [16]. In this sense, Al-, Zr-, and Hf-based catalysts have shown excellent catalytic performance in the CTH reaction of furfural to obtain FOL [17,18,19,20,21,22,23,24,25]. In many cases, the activity is so high that the reaction is not retained in FOL since consecutive reactions, as a consequence of the coexistence of Lewis and Brönsted acid sites, take place [26,27,28]. This has led to selectivity patterns with a broad spectrum of products, such as alkyl furfuryl ether, alkyl levulinates, or γ-valerolactone, which can be used as fuel additives, solvents or starting molecules for the synthesis of other valuable products [23,29,30,31,32]. However, these consecutive reactions also hinder reaching high yields in a particular target product, and the formation of soluble and insoluble (humins) polymers has been observed due to the high tendency of FUR and FOL to polymerize [25,33].

In the present study, Nb-based catalysts have been synthesized, characterized, and evaluated for the valorization of furfural through CTH processes. Previous studies have reported that the dispersed NbO_x_ species in catalysts provide acid sites, in such a way that these catalysts can be used in dehydration, esterification, transesterification, or alkylation reactions, among others [34,35,36,37,38,39,40]. In all of them, the presence of Lewis and/or Brönsted acidity played a key role in reaching high conversion values of a particular product [41,42]. Taking these premises into account, Nb-based catalysts could have the potential for furfural reduction through CTH processes. Thus, several porous silicas with different textural properties (pore size and channel length) have been synthesized, and subsequently, niobium species were incorporated by incipient wetness impregnation. The resulting catalysts were tested in the one-pot FUR valorization, using 2-propanol as both H-donor and solvent in CTH processes.

## 2. Results and Discussion

### 2.1. Characterization of the Catalysts

In order to facilitate the reading and understanding of this study, the acronyms used to refer to the different catalysts are summarized in Table 1.

The evaluation of the long-range ordering of the supported Nb catalysts was performed by small-angle X-ray scattering (SAXS) analysis (Figure 1). In all cases, a broadband, whose maxima and intensity differ as a function of the synthesis methodology selected, was observed. Thus, those catalysts synthesized at room temperature (RT), without a pore-expanding agent, show the band associated to (*d*_100_) planes at about 2θ: 1.07°, which is shifted to lower 2θ values (0.92°) when the catalysts were synthesized under hydrothermal conditions, regardless the use or not of a pore expander, confirming pore width expansion of SBA-15 after aging at high temperatures [43]. The incorporation of a pore-expanding agent leads to a broader and less intense *d*_100_ reflection. From these data, it can be inferred that the use of benzene causes a pore expansion, although concomitantly the ordering of the porous structure decreases, in agreement with data published in the literature [44].

The presence of crystalline phases was evaluated by X-ray diffraction (Appendix A). In all cases, diffractograms only display a broad band with a maximum located at 2θ of 23.13°, which is typical of amorphous silica. On the other hand, diffraction peaks attributed to Nb species have not been observed, so the NbO_x_ incorporated into the porous silica must be highly dispersed and/or amorphous.

The morphology of the Nb-doped porous silicas was analyzed by TEM. Firstly, the study was carried out at low magnification (scale of 200 nm) (Appendix A). The porous SBA-15 silica possesses a worm-like structure, where parallel microchannels cross this structure longitudinally. In the present study, it can be observed how the length of the channels increases when SBA-15 is synthesized under hydrothermal conditions (Nb-Si-HT). The incorporation of fluoride ions in the synthesis step limits the growing of the porous silica, in such a way that the channel length is shorter in comparison to conventional SBA-15 silica [45,46]. On the other hand, the addition of a pore expander agent, such as benzene, during the synthesis provokes a drastic disorder of particles, leading to a more disordered structure. The study at higher magnification (scale of 50 nm) (Figure 2) evidences the honeycomb structure of the SBA-15 silica synthesized at room temperature (Nb-Si-RT) and under hydrothermal conditions (Nb-Si-HT). This structure is maintained when fluoride ions are added in the synthesis step (Nb-Si-FRT and Nb-Si-FHT), so it can be inferred that the addition of fluoride only limits the growth of the porous silica nanoparticles. However, the incorporation of benzene causes the disappearance of the honeycomb structure, because an increase in the width of the mesochannels would limit their interconnection with adjacent channels, so the addition of the pore-expanding agent and fluoride ions leads to a mesocellular foam structure.

The study of the dispersion of NbO_x_ species was performed by EDX using EDX mapping (Appendix A). In all cases, the S-TEM micrographs rule out the agglomeration of NbO_x_, which is also corroborated by the absence of the diffraction peaks ascribed to crystalline NbO_x_ species (Appendix A).

Textural properties of Nb-doped porous silicas were determined from their N_2_ adsorption-desorption isotherms at −196 °C (Figure 3). According to the IUPAC classification, the adsorption-desorption isotherms of porous silicas synthesized at room temperature, or under hydrothermal conditions, can be considered as Type IV, which is typical of mesoporous materials [47]. However, the addition of fluoride ions and pore-expander (Nb-Si-BFHT), under hydrothermal conditions, leads to a Type II isotherm, characteristic of macroporous materials [47]. The isotherm of Nb-based catalyst supported on commercial silica is also considered as Type II, with macropores associated with voids between the spherical silica particles. When the synthesis is modified with fluoride species and, mainly, with the addition of a pore-expander agent, the wide of the hysteresis loop increases, suggesting a widening of the pore distribution. Thus, the hysteresis loops of Nb-Si-RT and Nb-Si-FRT are considered as H1, which is typical of uniform mesopores, which is due to the use of P-123 as a template in the synthesis generates uniform mesochannels. For Nb-Si-BFRT, the hysteresis loop is broader, being considered as H_2_, due to the pore-blocking/percolation in a narrow range of pore necks, or to cavitation-induced evaporation [47]. The synthesis of the porous silicas under hydrothermal conditions causes a shift of the hysteresis loops at higher relative pressure, which would indicate that the pore diameter must be higher.

Regarding the specific surface areas determined using the BET theory [48], these vary between 342 and 490 m^2^/g (Table 2). However, pore volumes show clear differences depending on the synthesis method. Thus, hydrothermal conditions allow a greater pore volume to be achieved than using room temperature., attaining the maximum pore volume for Nb-Si-BFHT (1.470 cm^3^/g). In this sense, previous works have reported that the use of a lower aging temperature promotes a stronger interaction between adjacent P-123 micelles, whereas higher temperatures favor the distancing of the P-123 micelles, so the porosity increases after calcination [43,49].

The analysis of the microporosity by the *t*-plot method, using the de Boer equation [50] (Table 2), reveals that the synthesis at room temperature leads to a higher proportion of micropores. In this sense, it has been reported that the microporosity of SBA-15 silica is ascribed to the interconnection of parallel mesochannels [51]. However, the increase in the aging temperature causes an adverse effect on microporosity, since, as previously noted, the P-123 micelles are more isolated, so the microporosity diminishes after the removal of the organic moieties [49]. On the other hand, it is worth highlighting the high microporosity of Nb-Si-BFRT sample (164 m^2^/g from *t*-plot:and a micropore volume of 0.072 cm^3^/g), which could be ascribed to the increase in the dimensions of the template, which also favors the interconnection of these parallel micelles in the aging step at room temperature, so a higher proportion of micropores are formed after the calcination step.

The pore size distribution was determined by Density Functional Theory (DFT) (Figure 4A) [52]. Nb-Si-RT displays a maximum at about 36 Å. The narrow and small pore diameter is directly related to the small hysteresis loop (Figure 3A) [47]. The use of a higher temperature in the synthesis (Nb-Si-HT) favors the increase in the pore diameter (50 Å) [43]. The addition of a pore expander agent and fluoride, which limits the growth of silica nanoparticles (Nb-Si-FRT, Nb-Si-FHT, or Nb-Si-BFRT), leads to catalysts with higher pore diameters, whose maxima are between 86 and 91 Å, as they show a shift of the hysteresis loop at higher relative pressure (Figure 3A,B). It is noteworthy that it is not possible to determine the pore size distribution for Nb-Si-BFHT and Nb-SiO_2_ catalysts due to the pore diameter must be out of the range of the DFT method, as evidence of their hysteresis loops, which are shifted to relative pressure close to P/P_0_ = 1.

As SBA-15 is a porous silica with a high microporosity due to the interconnection between adjacent mesochannels, the micropore size distribution was also evaluated by the MP method (Figure 4B) from their respective N_2_ adsorption-desorption isotherms (Figure 3A,B) [53]. In this sense, previous studies have reported that the microporosity is directly related to the N_2_ adsorbed at low relative pressure [47]. Both Nb-Si-RT and Nb-Si-HT possess micropores of 0.6 nm. In spite of the hydrothermal treatment must cause a decrease in the connection between mesochannels and, therefore, a decay of the microporosity, as can be seen in Table 2, Figure 4B reveals the presence of micropores with a pore diameter of 0.7 nm. The incorporation of structure-modifying agents, such as fluoride or benzene, favors the formation of larger micropores (1.0–1.3 nm). Finally, Si-BFHT and Nb-SiO_2_ display the largest micropores with maxima at 1.7 and 1.8 nm, respectively.

In order to analyze the surface chemical composition, a XPS study was carried out (Table 3). All the catalysts display similar surface atomic concentrations, with a Nb content between 0.48 and 0.74 wt%. Regarding the detailed study of each region, C 1s core level spectra show a band at a binding energy (BE) of 284.6 eV, assigned to adventitious carbon, which is also used as a reference for charge effect correction. A single contribution is also observed for Si 2p, at BE values of 103.4–103.7 eV, ascribed to Si in the form of SiO_2_. On the other hand, the analysis of the O 1s core level spectra shows the presence of two contributions (Figure 5A): a main contribution (about 97% of the total signal) located at 532.8–533.0 eV, typical of oxygen in silica, and a minor contribution (about 3%) at about 530.5–530.8 eV, associated with oxygen in NbO_x_ species. Finally, the Nb 3d core level spectra (Figure 5B) show the typical doublet of the Nb 3d_5/2_ and Nb 3d_3/2_, where Nb 3d_5/2_ signals appear at about 207.7–207.9 eV, assigned to Nb_2_O_5_ species.

Previous studies have reported that the one-pot conversion of FUR into valuable chemicals requires the presence of acid sites [26,27,28]. Thus, the total acidity has been determined by NH_3_-TPD (Table 2), affording a total amount of acid sites between 138 and 346 µmol/g. It should be emphasized that the catalyst with the narrowest pore diameter and longest channels, i.e., Nb-Si-RT, is also that with a lower proportion of acid sites. This fact could be ascribed to the stacking of NbO_x_ species, which can block access to long mesochannels. The increase in the pore diameter, or the shortening of the channel, seems to improve the acidity, probably due to a higher dispersion of the NbO_x_ species. All NH_3_-TPD profiles (Appendix A) are similar, with maximum desorption of ammonia molecules at about 210–220 °C, so the strength of the acid sites is weak, being similar in all the catalysts.

In order to elucidate the nature of acid sites (Lewis and/or Brönsted), pyridine adsorption coupled with FT-IR spectroscopy was employed (Figure 6). 

The absorption band located at 1540–1550 cm^−1^ is ascribed to protonated pyridine, i.e., pyridine adsorbed onto Brönsted acid sites, whereas pyridine adsorbed onto Lewis acid sites provides a band at 1440–1450 cm^−1^. The vibration band at 1490 cm^−1^ is associated with pyridine adsorbed on both Lewis and Brönsted acid sites, while those at 1575 and 1590 cm^−1^ are also ascribed to the presence of Lewis acid sites, although both bands are less intense than that observed at 1440 cm^−1^ [54]. Considering these assignations and according to the data reported in Figure 6, all catalysts mainly present Lewis-type acidity. since the bands located at 1440 cm^−1^ predominate, in such a way that the percentage of Lewis acid sites is 69.9–84.4%. Temperature-programmed desorption studies reveal that most of the pyridine is desorbed at around 200 °C, so the acid sites are weak. With regard to the quantification of the total amount of Brönsted and Lewis acid sites (Table 4), the values differ from those obtained from NH_3_-TPD, which can be explained by the different basicity and dimensions of NH_3_ and pyridine molecules, being easy for NH_3_ the access to micropores. Nonetheless, the catalyst with higher NH_3_-TPD values, i.e., Nb-Si-FRT, is also the catalyst that showed the highest pyridine desorption values. On the opposite, Nb-SiO_2_ showed the lowest values by NH_3_-TPD and pyridine desorption.

### 2.2. Catalytic Results

Once Nb-based materials were characterized, these were tested in the FUR hydrogenation to obtain valuable products in a one-pot process (Figure 1), using alcohol (iso-propanol) as a hydrogen donor, and the catalytic data are shown in Figure 7.

FUR conversion increases with the reaction time (Figure 7A), obtaining high values from 6 h of reaction. A clear influence of the textural properties of Nb-based catalysts can also be inferred, because the chemical composition is similar in all cases. Thus, those catalysts synthesized at room temperature reach high conversion values at shorter reaction times, which could be ascribed to the presence of narrower pores favoring an intimate contact between FUR and alcohol molecules with active sites (Figure 4 and Table 2), speeding up the reaction.

Among the samples synthesized at room temperature, Nb-Si-FRT reached the highest conversion at a shorter time (93% conversion after 6 h at 170 °C). In this sense, the shortening of the channels by the presence of fluoride species in the synthesis of the porous silica seems to minimize diffusional problems of the reactive and products, as well as the partial blockage of small mesochannels, since both FUR and FOL are highly reactive molecules, which tend to suffer polymerization reactions leading to the formation of carbonaceous deposits on the catalyst surface, causing deactivation [33]. This catalyst also presents a higher dispersion of the NbO_x_ species and, consequently, a higher amount of acid sites, as can be deduced from XPS (Table 3) and acidity data (Table 2) due to its better textural properties with shorter mesoporous channels. On the other hand, the use of a pore-expander agent, such as benzene, slightly worsens FUR conversion, confirming that larger pores do not exert a beneficial effect on the catalytic behavior. In the case of the Nb-based catalysts synthesized under hydrothermal conditions with higher pore diameters, the conversion values are lower than those attained with similar catalysts synthesized at room temperature. Thus, the catalyst with the highest pore diameter, i.e., Nb-Si-BFHT sample, displayed the poorest FUR conversion, about 70% after 6 h at 170 °C. These data would confirm that both the pore size and length play an important role in the catalytic behavior, achieving the best results for those catalysts with narrow and short pores.

Regarding the selectivity pattern, in all cases, furfuryl alcohol (FOL) and isopropyl furfuryl ether (ipFE) are the main products (Figure 7B,C). Both products are interesting, since FOL is largely used in the polymer field [8], while IpFE is considered a fuel additive because it increases the cetane index [55]. Strikingly, the yields of FOL and IpFE decrease over time, which would suggest the existence of consecutive reactions. Thus, FOL yield is higher between 30 and 90 min, reaching a maximum value of 20% after 90 min at 170 °C. However, the highest IpFE yield is attained after a longer reaction time (12 h at 170 °C) with a maximum value close to 70% with Nb-Si-FRT. These data agree with the literature, since FUR is firstly converted into FOL through catalytic transfer hydrogenation between furfural and i-propanol, via a six-membered intermediate formed on Lewis acid sites [16]. In this step, the role of the solvent/sacrificing alcohol is a key parameter, reaching the highest values when 2-propanol or 2-butanol were used [17,56]. In the next step, FOL is etherified with the alcohol, used as a solvent, to form IpFE, requiring Lewis or Brönsted acid sites [26]. In this sense, it has been reported that other catalysts, such as Zr-based catalysts or zeolites, do not stop the reaction in FOL or IpFE, evolving towards other consecutive products in the one-pot process, in such a way that the IpFE yield with these catalysts is relatively low in comparison to the catalysts reported in this study [22,26,57]. Thus, it can be concluded that the use of Nb-based catalysts would favor the formation of alkylfurfuryl ethers.

All these Nb-based catalysts also produce isopropyl levulinate (IpL) from furfural (Figure 7D), which is formed by the hydration of FOL and/or IpFE over Brönsted acid sites [26]. This fact was confirmed from the yield profiles, since the decay of FOL and IpFE yields is accompanied by an increase in the IpL yield, although this process is very slow, since the maximum IpL yield is only 18% after 24 h at 170 °C. Therefore, the high IpFE yield, together with the relatively low presence of IpL, would confirm that the amount of Brönsted acid sites is lower in comparison to Lewis ones, as was inferred from the pyridine-adsorption studies (Figure 6 and Table 4) [26].

The catalytic transfer hydrogenation of IpL with the sacrificing alcohol, i.e., i-propanol, gives rise to the reduction of the carbonyl group, obtaining i-propyl 4-hydroxy pentanoate as a product, which is catalyzed by Lewis acid sites [26]. However, this product was not detected because it undergoes a fast lactonization reaction to form γ-valerolactone (GVL) on Lewis or Brönsted acid sites [26]. Nevertheless, in all Nb-based catalysts, the GVL yield is very low, with a maximum of about 5% after 24 h at 170 °C (Figure 7E). These low yields are expected since the reaction appears to be retained in IpFE, and the amount of products formed from it is more limited. On the other hand, other authors have carried out kinetic studies concluding that the lactonization process for the formation of GVL requires longer times and more severe temperatures compared to the other reactions involved in the one-pot process of FUR conversion [58].

In addition, other products have been detected, although in minor proportions. For example, the aldol condensation between FUR and acetone, obtained by oxidation of the sacrificing alcohol, 2-propanol, in the CTH reaction, leads to 4-(2-Furyl)-3-buten-2-one (FBO) (Figure 7F) [16], although, in all cases, its yield was less than 2%. On the other hand, only small yields of angelica lactone (AL) were attained (Figure 7G), because this product is not thermodynamically favored, since it is only formed under vacuum conditions [59].

The amount of non-detected (ND) products (Figure 7H) increases along the reaction time, which is directly related to the existence of undesired reactions because of the polymerization of FUR or FOL in the presence of acid catalysts [33,60]. Among the Nb-based catalysts, those with longer channels (Nb-Si-RT and Nb-Si-HT) also display a higher proportion of non-detected products, attaining a maximum value of 37% after 24 h at 170 °C. The length of these channels can favor their partial blockage, causing diffusional problems. The addition of fluoride, which diminishes the length of the channels, or the use of a pore-expander agent, provokes fewer limitations for the diffusion of reactants and products and, therefore, the formation of a lower fraction of non-detected products.

Later, the effect of the reaction temperature was evaluated (Figure 8), and the catalytic data showed that conversion increases with the temperature. In the same way, the textural properties play a key role, inasmuch as those catalysts with narrower pore diameters and shorter pore lengths reached higher conversion values throughout the range of temperatures tested, being Nb-Si-FRT the most active catalyst. Regarding the obtained products, all catalysts present a similar selectivity pattern, although their yields depend on conversion values. This is due to all the catalysts displaying similar types and strengths of acid sites, being the only difference associated with their textural properties. Thus, at low temperatures, FOL is formed on Lewis acid sites. However, an increase in the temperature causes a progressive decrease in FOL, which is accompanied by the formation of a higher amount of IpFE, in such a way that the increase in temperature promotes the etherification reaction. The use of the highest reaction temperature favors the rehydration of the furan ring of IpFE, forming a small amount of IpL, which is obtained via Brönsted acidity, although its concentration is low, as inferred from pyridine adsorption coupled to FT-IR spectroscopy, so the one-pot reaction barely evolves from IpFE (Figure 8 and Table 4). From these data, it can be concluded that Nb-based catalysts are highly selective towards IpFE, mainly at higher temperatures, due to the low proportion of Brönsted acid sites, thus preventing the formation of more advanced products in the one-pot reaction of FUR, reaching yields close to 65% after 6 h at 170 °C.

An important issue related to heterogeneous catalysis is the physicochemical characterization of the catalysts used to detect any modification with respect to the fresh one. Thus, the catalysts used in the catalytic process were recovered after 6 h of reaction at 170 °C to be characterized by N_2_ sorption at −196 °C and XPS. These spent catalysts were labeled with the final letter -u.

The analysis of the textural properties (Table 5) evidences a clear decrease in the S_BET_ values and mainly of microporosity, as deduced from the *t*-plot. This trend is more pronounced in the case of the catalysts with a higher microporosity, as inferred from their pore size distribution obtained by DFT and MP methods (Figure 9 and Appendix A). These data suggest that micropores can be partially blocked along the reaction time, diminishing the amount of available active sites involved in the one-pot process.

The analysis of the chemical composition of the used catalysts was carried out by XPS (Table 6). In all cases, an increase in the C content on the catalyst surface can be observed, confirming the formation of carbonaceous deposits, probably due to the strong interaction of highly reactive molecules involved in this reaction, such as FUR or FOL. This would imply a decrease in the Si and Nb contents (Figure 10A). The C 1s core level spectra are made up of four contributions: (i) the main contribution located at 284.8 eV is attributed to C-C bonds, (ii) at 286.2 eV is assigned to the coexistence of C-O-C and C-O-H bonds; (iii) the band centered at 287.3 eV is attributed to -C=O bonds, and (iv) at 289.2 eV is associated to -O-C=O (Figure 10B). The presence of these bands in the C 1s spectrum would confirm the deposition of organic species formed from FUR and/or FOL on the catalyst surface.

In the next step of the catalytic study, the most active catalyst, i.e., Nb-Si-FRT, was compared with a catalyst with a similar Nb loading but supported on a commercial SiO_2_ (Figure 11). The data obtained reveal that the latter catalyst (Nb-SiO_2_) displays a poorer catalytic activity, since it does not reach a full conversion after 24 h at 170 °C. This difference is more pronounced at shorter reaction times. However, the selectivity patterns are similar, showing the coexistence of FOL and IpFE at a shorter reaction time. The progressive decrease in FOL yield and the concomitant increase in IpFE would confirm that IpFE is obtained from the etherification of FOL. As previously stated, these catalysts show a higher proportion of Lewis acid sites compared to Brönsted ones, which would imply that the etherification of FOL to obtain IpFE must take place on Lewis acid sites [26]. Because of the low proportion of Brönsted acid sites, the formation of IpL is limited, since this product requires longer reaction times, being its yields relatively low in comparison to other catalysts, such as Zr-based catalysts [58,60]. On the other hand, it is also noteworthy that both catalysts display a similar proportion of non-detected products at different reaction times. Therefore, as previously noted, it could be expected that the improvement in the catalytic behavior of the Nb-Si-FRT catalyst must be ascribed to its textural properties, which maximize the number of available acid sites (Table 2 and Table 4). In this sense, TEM micrographs have revealed that Nb_2_O_5_ is well dispersed in both Nb-Si-RFT and Nb-SiO_2_ catalysts; however, Nb_2_O_5_ particles enter inside the porous structure in Nb-Si-RTF, while remaining on its surface in the case of Nb-SiO_2_.

In order to understand the mechanism and identify the limiting steps of the one-pot reaction, different intermediates (FOL and IpL) have been fed into the reactor instead of furfural (Figure 12). Starting from FOL, the conversion is similar to that reached with FUR, although the yields of the different products differ. Thus, IpFE is the main product with a yield of 50% after 12 h at 170 °C, which is lower than that attained when FUR was used. In addition, it is also noteworthy that the decay of IpFE is accompanied by an increase in the IpL yield (27%), so this data would suggest that the use of FOL as a reactant allows reaching more advanced reaction products in the one-pot process compared to FUR, although the existence of a lower proportion of Brönsted acid sites would limit the formation of IpL. In this sense, FUR is a molecule prone to undergoing polymerization, so this fact makes its progress in the one-pot process difficult since the active sites involved in the reaction could be deactivated. However, FOL is also another molecule that can suffer polymerization, and, in fact, it is used in the polymer industry. Nonetheless, the catalytic results show how the amount of non-detected products is lower when the reaction starts from FOL.

When IpL is used as feedstock, the conversion values are much lower compared to FUR and FOL, so it seems that its lactonization to form GVL is a thermodynamically limited step in the one-pot process, which agrees with previous studies with Zr-based catalysts [58,60]. On the other hand, it is also noteworthy that the amount of non-detected products is more limited in comparison to the other reactants, confirming that FUR and FOL are highly reactive and responsible for the formation of carbonaceous deposits on the catalyst surface.

An important advantage of heterogeneous catalysts is their ability to be reused over several cycles. Thus, the reusability of the Nb-Si-FRT catalyst was also evaluated (Figure 13). The catalytic data reveal that FUR conversion decreases with the number of cycles (from 92 to 75% in the third cycle), probably due to the formation of carbonaceous deposits on the catalyst surface, causing a partial blockage of active sites necessary for the one-pot process, as inferred from the characterization of the used catalyst (Table 5 and Table 6). In addition, the selectivity patterns differ as the number of cycles increases, since GVL and mainly IpL are observed in the first cycle. However, both products decrease after subsequent runs, not being detected in the third cycle. Similarly, the IpFE yield also decreases from 64% after the first cycle to 43% after the third cycle, at 170 °C for 3 h. In contrast, FOL is the unique product that increases its yield after the reuse of the Nb-Si-FRT catalyst, from 3% after the first cycle to 18% after the third cycle. From these data, it can be concluded that the reuse of the catalyst causes a progressive deactivation of the catalyst, and the one-pot process is retained in earlier steps since, after the first catalytic cycle, advanced reaction products, such as GVL and IpL, were detected, while, after the third cycle, only the presence of IpFE and FOL were identified.

## 3. Materials and Methods

### 3.1. Reagents

The synthesis of Nb-based catalysts supported on porous silica was performed using Pluronic P-123 (PEO20PEO70PEO20, average Mn of 5800 g/mol, Sigma-Aldrich, Saint Louis, MO, USA), as a template, hydrochloric acid (37%, VWR, Radnor PA, USA) and tetraethylorthosilicate (98%, Sigma-Aldrich). The pore expander used in this study was benzene (99%, Sigma-Aldrich), while the fluoride source used to shorten the channels was ammonium fluoride (99.5%, Sigma-Aldrich). The Nb source was niobium oxalate (98%, Alfa Aesar, Ward Hill, MA, USA). In an additional study, a commercial fumed silica (Aldrich) with a surface area of 200 m^2^/g was employed for comparison.

Chemicals used in the one-pot processes were furfural (99%, Sigma-Aldrich), 2-propanol (99.9%, HPLC grade, VWR) as sacrificing alcohol, and o-xylene (99.9%, Sigma-Aldrich) as internal standard.

The gases required for analysis and characterization were: He (99.99%, Air Liquide, Paris, France), H_2_ (99.999%, Air Liquide), N_2_ (99.9999%, Air Liquide), N_2_/O_2_ (80/20 vol %, 99.99%, Air Liquide) and NH_3_/He (5% vol in NH_3_, 99.99%, Air Liquide).

### 3.2. Synthesis of Catalysts

The synthesis of porous silicas with different textural properties was carried out following the methodology previously described by Lopez-Asensio et al. [33].

For the synthesis of porous silica without structural modifications, P123 was dissolved in a solution of HCl (1.7 M), and then, tetraethylorthosilicate (TEOS) was added dropwise, obtaining a gel with the following molar composition: 1 P123: 55 SiO_2_: 350 HCl: 11,100 H_2_O. Next, this gel was aged for 72 h at room temperature, in the case of the Nb-Si-RT sample, whereas the Nb-Si-HT sample was maintained for 24 h at room temperature and, then, 48 h in a Teflon-lined autoclave at 100 °C.

For the synthesis in the presence of fluoride species, used to shorten the channel length, P-123 and NH_4_F were dissolved in a solution of HCl (1.7 M). Then, TEOS was added dropwise to obtain a gel with a molar composition of 1 P123: 55 SiO_2_: 350 HCl: 1.8 NH_4_F: 11,100 H_2_O. Similar to the previous syntheses, the gel was aged for 72 h at room temperature, 24 h at room temperature, and then 48 h in a Teflon-lined autoclave at 100 °C for Nb-Si-FRT and Nb-Si-FHT samples, respectively.

Finally, benzene was added to increase the pore diameter of the samples. In this type of synthesis, both P-123 and NH_4_F were dissolved in a solution of HCl (1.7 M), and then the pore expander (benzene) was incorporated. After 30 min, TEOS was added dropwise to obtain a molar composition of 1 P123: 55 SiO_2_: 48 B: 350 HCl: 1.8 NH_4_F: 11100 H_2_O. Nb-Si-BFRT and Nb-Si-BFHT samples were obtained from gel aged by following similar methodologies to those previously described.

After aging, gels were filtered, washed with distilled water, and dried at 80 °C overnight. Then, solids were calcined at 550 °C, with a heating rate of 1 °C/min, for 6 h, to obtain several porous silicas with different textural properties.

The incorporation of Nb species was carried out by incipient wetness impregnation using niobium oxalate. In all cases, an aqueous solution of niobium oxalate was dissolved in a solution of oxalic acid (0.1 M) to incorporate an amount of 12 wt.% of Nb_2_O_5_ on each support. After the impregnation, samples were dried at 80 °C and calcined at 400 °C for 2 h, using a heating rate of 2 °C/min. In order to compare the obtained results with a commercial fumed silica, this support was impregnated with 12 wt.% of Nb_2_O_5_, following the methodology described above. Then, the sample was dried at 80 °C overnight, and finally, the precursor was calcined at 400 °C for 2 h with a ramp of 2 °C/min.

### 3.3. Characterization Techniques

Crystallinity and long-range ordering were determined by X-ray diffraction (XRD) and small-angle X-ray scattering (SAXS), respectively. The morphology of porous silicas was studied by Transmission Electron Microscopy (TEM). Textural properties were evaluated from their N_2_ adsorption-desorption isotherms at −196 °C. The analysis of acidity was carried out by temperature-programmed desorption of NH_3_ (NH_3_-TPD) and pyridine adsorption coupled to FT-IR spectroscopy. The surface chemical composition of the Nb-doped porous silicas was estimated by X-ray photoelectron spectroscopy (XPS). A detailed description of each technique is provided as Appendix A.

### 3.4. Catalytic Tests

All catalytic tests were carried out in glass pressure reactors with thread bushing (Ace, 15 mL). In each test, 100 mg of catalyst was mixed with FUR (0.1 mmol) and 2-propanol (5 mL), maintaining a 2-propanol/FUR molar ratio of 50:1. Prior to each test, glass reactors were purged with helium. Reactions were performed under continuous stirring (400 rpm), in a range of temperature between 110 and 170 °C, at different reaction times. An aluminum block was used for reactions, controlling temperature with a thermocouple. After the reaction, the glass reactors were moved away from the aluminum block to water to cool down the reaction medium and filtered to take an aliquot.

In the case of the reusability studies, once the reaction was finished, the reactors were cooled down, and an aliquot of the reaction solution was extracted by decantation to be analyzed by GC. Then, the catalyst was filtered and washed with the solvent used in the reaction (2-propanol). Finally, the catalyst was dried before performing the next catalytic cycle.

The aliquots were analyzed by gas chromatography using a flame ionization detector and a CP-Wax 52 CB capillary column.

The FUR conversion and yield were calculated as follows:$$Conversion(\%)=\frac{\mathrm{mol}\mathrm{of}\mathrm{furfural}\mathrm{converted}}{\mathrm{mol}\mathrm{of}\mathrm{furfural}\mathrm{fed}}\times 100$$
$$Yield\left(\%\right)=\frac{\mathrm{mol}\mathrm{of}\mathrm{product}}{\mathrm{mol}\mathrm{of}\mathrm{furfural}\mathrm{fed}}\times 100$$

## 4. Conclusions

Several porous silicas with different textural properties have been synthesized to support NbO_x_ species. The analysis of the textural properties revealed that the temperature of synthesis, the addition of a pore-expander agent, or fluoride ions, affects the porosity of the silicas seriously.

The characterization of the Nb-based catalysts has demonstrated that the incorporation of Nb species promotes the formation of acid centers, which are mainly of the Lewis type, although the strength of these acid centers is low-medium.

The presence of these acid sites makes these catalysts interesting for catalytic transfer hydrogenation processes to obtain high value-added chemical in one pot. The catalytic results show a similar selectivity pattern in all cases, since FOL is obtained at shorter times due to the hydrogen transfer from isopropanol to FUR through Lewis acid sites, whereas IpFE is formed at longer reaction times, because of the etherification of FOL with isopropanol. The presence of a lower proportion of Brönsted acid sites limits the formation of IpL and other more advanced products in the one-pot reaction.

Concerning the influence of the textural properties on the catalytic behavior, the catalytic results reveal that the presence of shorter channels and narrow pore diameter favors the higher conversion of FUR at shorter reaction times.

## Data Availability

The data presented in this study are available on request from the corresponding author.

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
