# Peer review of "Nb-Based Catalysts for the Valorization of Furfural into Valuable Product through in One-Pot Reaction"

_ijms, 2024, doi:10.3390/ijms25052620_

Round 1

Reviewer 1 Report

Comments and Suggestions for Authors

This study presents the synthesis of NbOx-incorporated mesoporous silica for furfural valorization. While the results and discussion contribute scientifically, there is a need for substantial innovation. The paper also exhibits deficiencies in its writing. A major revision is necessary before being accepted for publication, addressing the following comments:

1. What distinguishes Nb-based catalysts in furfural conversion, particularly in terms of activity, selectivity, and stability compared to other catalysts? Although the authors discuss Lewis and/or Bronsted acidity, is the activation ability of NbOx for furfural related to its electronic structure? Could NbOx play a role in promoting or stabilizing key intermediates of the target products?

2. What is the optimized catalyst design? The manuscript outlines the catalyst screening process but lacks a clear catalyst design concept.

3. The study's highlight is unclear, failing to adequately convey the catalyst's superiority. Mechanism explanations for performance variations under different catalysts and reaction conditions are often speculative.

4. The manuscript extensively covers porous structure and acidic site characterizations, but the structure-activity relationship of the catalyst remains under-analyzed. It is suggested to condense characterization results and present them more concisely.

5. In Figure 5, variations in O1s and Nb 3d XPS peak positions suggest potential differences in chemical states among samples. This needs verification and clarification.

6. Avoid using phrases like "it is well known..." (Lines 203, 410, 427) as not all readers may be familiar with catalytic furfural conversion.

Author Response

Reviewer 1

This study presents the synthesis of NbOx-incorporated mesoporous silica for furfural valorization. While the results and discussion contribute scientifically, there is a need for substantial innovation. The paper also exhibits deficiencies in its writing. A major revision is necessary before being accepted for publication, addressing the following comments:

  1. What distinguishes Nb-based catalysts in furfural conversion, particularly in terms of activity, selectivity, and stability compared to other catalysts? Although the authors discuss Lewis and/or Bronsted acidity, is the activation ability of NbOx for furfural related to its electronic structure? Could NbOx play a role in promoting or stabilizing key intermediates of the target products?

The authors appreciate the reviewer's comment, and the issues raised will be clarified. As it is indicated in the Introduction section, furfural is converted in furfuryl alcohol by the participation of Lewis acid sites. Previous studies have reported that the presence of Lewis acid sites in Al-, Zr- or Hf-based catalysts promotes the CTH process. Furthermore, this reaction takes places through a six-membered intermediate, where a secondary alcohol (2-propanol) transfers a hydride to the aldehyde group of furfural leading to its reduction.

The mechanism proposed for this first step has been reported in several papers. Thus, in the following scheme, the CTH mechanism of furfural in the presence of a Lewis acid catalyst as Al2O3 is shown (https://doi.org/10.1016/j.apcata.2018.02.022).

However, the coexistence of Lewis and Brönsted acid sites can promote consecutive reactions. Thus, Scheme 1 included in the manuscript shows the type of acid sites involved in each step in the one pot process studied (pages 10 and 11).

Therefore, a key parameter for this study is the identification and quantification of the different types of acid sites, which have been carried out by pyridine adsorption coupled to FTIR spectroscopy. This study reveals a high proportion of Lewis acid sites, while the amount of Brönsted acid sites is lower. This implies that the reaction stops at the stages in which the resulting products require Brönsted acid sites for their transformation into others..

  1. What is the optimized catalyst design? The manuscript outlines the catalyst screening process but lacks a clear catalyst design concept.

The authors thank this comment. The percentage of NbOx (12 wt%) and the synthetic procedure were optimized in previous studies performed in the research group. In the present work, the influence on the catalytic performance of the silica support used for the preparation of these supported NbOx catalysts has been evaluated, as well as experimental variables such as reaction temperature and time, similar to other studies published in the literature.

  1. The study's highlight is unclear, failing to adequately convey the catalyst's superiority. Mechanism explanations for performance variations under different catalysts and reaction conditions are often speculative.

The authors thank the advice of the reviewer. In the last paragraph of the Introduction section, it is highlighted that this is the first study where Nb-based catalysts have been evaluated in the one-pot conversion of furfural to obtain valuable products. In the same way, the amount of Lewis and Brönsted acid sites plays an important role in the selectivity pattern, which can be also tuned by the choice of the porous silica used for the preparation of Nb-based catalysts, due to their different textural properties.

In this sense, the furfural conversion and selectivity pattern have been correlated with acidity and textural properties.

  1. The manuscript extensively covers porous structure and acidic site characterizations, but the structure-activity relationship of the catalyst remains under-analyzed. It is suggested to condense characterization results and present them more concisely.

Thanks for the consideration of the reviewer, the aim of the manuscript is to find a correlation between the textural properties and acidity with the catalytic activity and the novelty of the use of Nb-based catalysts.

The authors understand the suggestion of the reviewer. As the Nb-content is the same in all catalysts, the manuscript is focused on a detailed study of the textural properties. In fact, the textural properties were analyzed before and after the reaction, being observed a clear blockage of the pores after the reaction which was more pronounced in the case of the catalysts with a higher microporosity. Moreover, those catalysts with narrower pore diameter and shorter pore length reached higher conversion values, being Nb-Si-FRT the most active catalyst.

  1. In Figure 5, variations in O 1s and Nb 3d XPS peak positions suggest potential differences in chemical states among samples. This needs verification and clarification.

The authors have referred the binding energy against the adventitious carbon (284.8 eV). Considering that all support are silica and Nb-species have been incorporated by the same methodology, the shift of the binding energies must be negligible. The authors have deconvoluted each region, identifying and quantifying each contribution.

  1. Avoid using phrases like "it is well known..." (Lines 203, 410, 427) as not all readers may be familiar with catalytic furfural conversion.

Following the advice the reviewer, the term “it is well known….” has been removed from the manuscript (marked in yellow).

Reviewer 2 Report

Comments and Suggestions for Authors

In this manuscript, the author designed a series of Nb-based catalysts supported on porous silica with different textural properties for furfural conversion. Overall, some problems should be revised before the acceptance.

1.     Isopropyl furfuryl ether (IpFE) was the main product for furfural conversion over Nb-based catalysts. The solvent of isopropanol was important to produce IpFE. Thus, the effect of solvent should be investigated in this work. A paper about the solvent effect in the hydrogenation of biomass can be cited in this manuscript: Catalysis Today, 2023, 423: 114252.

2.     The conversion rate was not marked in the Figures about catalytic results.

3.     The authors proposed that Lewis acid sites were crucial for the IpFE production. If there is no Lewis acid site on the catalysts, will this product be generated? Both Lewis acid sites and Brönsted acid sites existed in Nb-based catalysts supported on porous silica. What is the role of Brönsted acid sites in furfural conversion?

4.     The catalytic stability of Nb-Si-RFT catalyst is not satisfactory. The author can try to remove carbon deposition to regenerate the catalytic performance.

Comments on the Quality of English Language

In this manuscript, the author designed a series of Nb-based catalysts supported on porous silica with different textural properties for furfural conversion. Overall, some problems should be revised before the acceptance.

1.     Isopropyl furfuryl ether (IpFE) was the main product for furfural conversion over Nb-based catalysts. The solvent of isopropanol was important to produce IpFE. Thus, the effect of solvent should be investigated in this work. A paper about the solvent effect in the hydrogenation of biomass can be cited in this manuscript: Catalysis Today, 2023, 423: 114252.

2.     The conversion rate was not marked in the Figures about catalytic results.

3.     The authors proposed that Lewis acid sites were crucial for the IpFE production. If there is no Lewis acid site on the catalysts, will this product be generated? Both Lewis acid sites and Brönsted acid sites existed in Nb-based catalysts supported on porous silica. What is the role of Brönsted acid sites in furfural conversion?

4.     The catalytic stability of Nb-Si-RFT catalyst is not satisfactory. The author can try to remove carbon deposition to regenerate the catalytic performance.

Author Response

Reviewer 2

Comments and Suggestions for Authors

In this manuscript, the author designed a series of Nb-based catalysts supported on porous silica with different textural properties for furfural conversion. Overall, some problems should be revised before the acceptance.

  1. Isopropyl furfuryl ether (IpFE) was the main product for furfural conversion over Nb-based catalysts. The solvent of isopropanol was important to produce IpFE. Thus, the effect of solvent should be investigated in this work. A paper about the solvent effect in the hydrogenation of biomass can be cited in this manuscript: Catalysis Today, 2023, 423: 114252.

The authors thank the suggestion of the Reviewer. In the present study, the reduction of furfural to furfuryl alcohol is carried out through the Meerwein-Ponndorf-Verley process. In this reaction, a secondary alcohol is used a sacrificing alcohol donating a hydrogen to the aldehyde or ketone to form the respective alcohol. This reaction takes place with the presence of Lewis acid sites through a six-membered intermediate. In previous studies, several secondary alcohols have been used in this reaction (2-propanol, 2-butanol and cyclohexanol) (Ref: https://doi.org/10.1016/j.apcata.2018.02.022). Among them, the best catalytic results were obtained for 2-butanol. However, the conversion using 2-propanol as sacrificing alcohol is very close to those obtained with 2-butanol, so 2-propanol was selected due to its lower cost.

Regarding to the formation of isopropyl furfuryl ether, this product is obtained by a etherification reaction of furfuryl alcohol through the Lewis or Brönsted acid sites, as it is reported in the literature (Ref: https://doi.org/10.1002/anie.201302575, https://doi.org/10.1039/D1SE00942G, https://doi.org/10.1016/j.jiec.2016.06.007).

Following the Reviewer’s recommendation, the authors have incorporated the reference proposed as Ref. 56.

  1. The conversion rate was not marked in the Figures about catalytic results.

The authors thank the comment of the Reviewer. The conversion values are shown in the y-axis. Considering the mass used of the catalyst (0.1 g) and the furfural fed (1 mmol), it is possible to determinate the rate, although the rate values will follow the same trend to that observed in the conversion of furfural, since all tests were carried out with the same mass and furfural fed.

  1. The authors proposed that Lewis acid sites were crucial for the IpFE production. If there is no Lewis acid site on the catalysts, will this product be generated? Both Lewis acid sites and Brönsted acid sites existed in Nb-based catalysts supported on porous silica. What is the role of Brönsted acid sites in furfural conversion?

We thank to the Reviewer the comment. As it has been exposed in the manuscript and explained to the previous Reviewer, Lewis acid sites are necessary to carry out the catalytic transfer hydrogenation of furfural to furfuryl alcohol. Then, once that furfuryl alcohol has been obtained, the etherification reaction with the alcohol can take place by the presence of Lewis or Brönsted acid sites.

In addition, Brönsted acid sites are required for the opening of the furan ring to obtain alkyl levulinates. However, the presence of a low proportion of Brönsted acid sites limits the formation of alkyl levulinate or g-valerolactone.

  1. The catalytic stability of Nb-Si-RFT catalyst is not satisfactory. The author can try to remove carbon deposition to regenerate the catalytic performance.

The results reported in the present study are similar to other previously reported in the literature. Both furfural and furfuryl alcohol are prone to suffer polymerization reactions forming humins, which are deposited on the catalyst surface causing a decrease in the available active sites.

In spite of this decrease in conversion, these heterogeneous catalysts can be regenerated after a thermal treatment at 550 ºC, since the carbonaceous deposits are calcined as it has been observed in other previous works (Essih et al. Adv. Sustainable Syst. 2022,6, 2100453).

Comments on the Quality of English Language

In this manuscript, the author designed a series of Nb-based catalysts supported on porous silica with different textural properties for furfural conversion. Overall, some problems should be revised before the acceptance.

  1. Isopropyl furfuryl ether (IpFE) was the main product for furfural conversion over Nb-based catalysts. The solvent of isopropanol was important to produce IpFE. Thus, the effect of solvent should be investigated in this work. A paper about the solvent effect in the hydrogenation of biomass can be cited in this manuscript: Catalysis Today, 2023, 423: 114252.

As was indicated previously, the role of the solvent is a key parameter in the catalytic transfer hydrogenation of furfural to furfuryl alcohol. The effect of the solvent was analyzed in previous studies, obtaining the best results when 2-propanol and 2-butanol were used as sacrificing alcohol. (Ref: https://doi.org/10.1016/j.apcata.2018.02.022).

  1. The conversion rate was not marked in the Figures about catalytic results.

The authors thank the comment of the Reviewer. The conversion values are shown in the y-axis. Considering the mass used of the catalyst (0.1 g) and the furfural fed (1 mmol), it is possible to determinate the rate, although the rate values will follow the same trend to that observed in the conversion of furfural, since all tests were carried out with the same mass and furfural fed.

  1. The authors proposed that Lewis acid sites were crucial for the IpFE production. If there is no Lewis acid site on the catalysts, will this product be generated? Both Lewis acid sites and Brönsted acid sites existed in Nb-based catalysts supported on porous silica. What is the role of Brönsted acid sites in furfural conversion?

We thank to the Reviewer the comment. As it has been exposed in the manuscript and explained to the previous Reviewer, Lewis acid sites are necessary to carry out the catalytic transfer hydrogenation of furfural to furfuryl alcohol. Then, once that furfuryl alcohol has been obtained, the etherification reaction with the alcohol can take place by the presence of Lewis or Brönsted acid sites.

In addition, Brönsted acid sites are required for the opening of the furan ring to obtain alkyl levulinates. However, the presence of a low proportion of Brönsted acid sites limits the formation of alkyl levulinate or g-valerolactone.

  1. The catalytic stability of Nb-Si-RFT catalyst is not satisfactory. The author can try to remove carbon deposition to regenerate the catalytic performance.

The results reported in the present study are similar to other previously reported in the literature. Both furfural and furfuryl alcohol are prone to suffer polymerization reactions forming humins, which are deposited on the catalyst surface causing a decrease in the available active sites.

In spite of this decrease in conversion, these heterogeneous catalysts can be regenerated after a thermal treatment at 550 ºC, since the carbonaceous deposits are calcined as it has been observed in other previous works (Essih et al. Adv. Sustainable Syst. 2022,6, 2100453).

Reviewer 3 Report

Comments and Suggestions for Authors

The article presents NB-based SIO2 catalyst designed for furfural conversion into more valuable products. The outcomes are clearly presented and sufficiently supported by data. Forthcoming notes should be clarified before the article acceptation:

1) Fig. 1 – the explanation of abbreviations (BFRT, FHT,  etc.) comes too late (after this figure, where they are used), should be noted earlier.

2) With which method is supported the argument, that the addition of fluoride limits only the growing of the porous silica nanoparticles? (page 3, end)

3) In TEM figures (Figure 2), the ruler/scale is not readable.

4) The argument, that Nb2O5 particles are highly dispersed in silica channels is not visible in figures with scale 200 nm (supporting info).

5) The paragraph about pore size distribution (page 6) should be better connected with the paragraph about adsorption isotherms, because the pore sizes are discussed there.

6) Table 1 – why is the trend at Nb-SI-RT / Nb-SI-HT in SBET different than at the other materials, despite the other parameters follows the same trend?

7) Fig. 4 – colours of the curves are not good distinguished. According to my opinion, the use of A is quite archaistic, conventionally, nm is used for pore diameter.

8) Which data shows, that Nb-SI-FRT presents a higher dispersion of the NbOx? (page 9, row 257).

9) The abbreviation GVL should be clarified somewhere in the text, not only in the Scheme 1.

10) Figure 8 + 9 – the legend clarification is missing – “ND”, “u”

11) The information, how the reused experiment was done is missing. What was the procedure between each step? Was the catalyst washed/dried? Were the losses of catalysts amount considered? This can be connected with the product found only in the second run but not in the third run.

12) Which data supports the mention, that Nb2O5 is well dispersed in Nb-SiO2?

Author Response

Reviewer 3

Comments and Suggestions for Authors

The article presents NB-based SIO2 catalyst designed for furfural conversion into more valuable products. The outcomes are clearly presented and sufficiently supported by data. Forthcoming notes should be clarified before the article acceptation:

1) Fig. 1 – the explanation of abbreviations (BFRT, FHT, etc.) comes too late (after this figure, where they are used), should be noted earlier.

The authors thank the suggestion of the Reviewer. In order to clarify the acronyms used in the present manuscript, a table (Table 1) with the acronyms has been added.

Table 1. List of the acronyms of the Nb-based catalysts.

Catalyst

Nb-Si-RT

Nb-doped mesoporous SBA-15 silica synthesized at room temperature

Nb-Si-HT

Nb-doped mesoporous SBA-15 silica synthesized under hydrothermal conditions (100 °C)

Nb-Si-FRT

Nb-doped mesocellular foam synthesized at room temperature

Nb-Si-FHT

Nb-doped mesocellular foam synthesized under hydrothermal conditions (100 °C)

Nb-Si-BFRT

Nb-doped mesocellular foam expanded with benzene synthesized at room temperature

Nb-Si-BFHT

Nb-doped mesocellular foam expanded with benzene synthesized under hydrothermal conditions (100 °C)

Nb-SiO2

Nb-doped commercial silica

2) With which method is supported the argument, that the addition of fluoride limits only the growing of the porous silica nanoparticles? (page 3, end)

It has been reported in the literature that the addition of fluoride species limits the growth of silica around the template. The authors have supported this sentence with two new references in the manuscript (Ref. 45: https://doi.org/10.1021/ja983218i and Ref. 46: https://doi.org/10.1021/jp065760w).

3) In TEM figures (Figure 2), the ruler/scale is not readable.

The authors agree with the Reviewer. However, the scale of the TEM micrographs is the same in all cases (50 nm), as indicated in the caption of Figure 2 and at the bottom of the micrographs.

4) The argument, that Nb2O5 particles are highly dispersed in silica channels is not visible in figures with scale 200 nm (supporting info).

The authors thank the advice of the reviewer. On one hand, diffraction peaks ascribed to Nb2O5 crystals have not been observed by XRD (Figure S1). On the other hand, the TEM micrographs do not show agglomeration of Nb-particles from the mapping of their S-TEM micrographs. Thus, the corresponding sentence has been modified as follows:

In all cases, the S-TEM micrographs rule out the agglomeration of Nb2O5, which is also corroborated by the absence of the diffraction peaks ascribed to crystalline NbOx species (Supplementary Information, Figures S1).”

5) The paragraph about pore size distribution (page 6) should be better connected with the paragraph about adsorption isotherms, because the pore sizes are discussed there.

The authors thank the suggestion of the Reviewer. An attempt has been made to correlate the pore size distribution, as well as the microporosity, with their respective N2 adsorption-desorption isotherms. Thus, the following parapgraphs have been modified in the manuscript:

The narrow and small pore diameter is directly related with the small hysteresis loop (Figure 3A) [47]. The use of a higher temperature in the synthesis (Nb-Si-HT) favors the increase in the pore diameter (50 Å) [43]. The addition of a pore expander agent and fluoride, which limits the growing of silica nanoparticles (Nb-Si-FRT, Nb-Si-FHT or Nb-Si-BFRT), leads to catalysts with higher pore diameter whose maxima are between 86 and 91 Å, as shows a shift of the hysteresis loop at higher relative pressure (Figure 3A and 3B). It is noteworthy that it is not possible to determine the pore size distribution for Nb-Si-BFHT and Nb-SiO2 catalysts due to the pore diameter must be out of the range of the DFT method, as evidence their hysteresis loops which are shifted to relative pressure close to P/P0=1.”

“As SBA-15 is a porous silica with a high microporosity due to the interconnection between adjacent mesochannels, the micropore size distribution was also evaluated by the MP method (Figure 4B) from their respective N2-adsorption isotherms (Figure 3A and 3B) [53]. In this sense, previous authors have reported that the microporosity is directly related to the N2 adsorbed at low relative pressure [47]. Both Nb-Si-RT and Nb-Si-HT possess micropores of 0.6 nm. In spite of the hydrothermal treatment must cause a decrease in the connection between mesochannels and, therefore, a decay of the microporosity, as can be seen in Table 2, Figure 4B reveals the presence of micropores with a pore diameter of 0.7 nm. The incorporation of structure-modifying agents, such as fluoride or benzene, favors the formation of larger micropores (1.0-1.3 nm). Finally, Si-BFHT and Nb-SiO2 display the largest micropores with maximum at 1.7 and 1.8 nm, respectively.”

6) Table 1 – why is the trend at Nb-SI-RT / Nb-SI-HT in SBET different than at the other materials, despite the other parameters follows the same trend?

Thank you for the question. The textural properties are different due to the addition of fluoride ions, which limits the growth of the silica along the template, while the use of the pore expander agent also modifies the textural properties.

7) Fig. 4 – colours of the curves are not good distinguished. According to my opinion, the use of A is quite archaistic, conventionally, nm is used for pore diameter.

Following the advice of the Reviewer, the Figures have been modified and x-axes have been changed to nm in all figures about the pore size distribution.

8) Which data shows, that Nb-SI-FRT presents a higher dispersion of the NbOx? (page 9, row 257).

The higher dispersion of the NbOx-species must be related to the textural properties. The presence of shorter channels must promote its dispersion. In the case of Nb-Si-RT, the use of longer channels should imply that Nb species should be located mainly on their external surface, or in the outermost parts of the pores, rather than along the channels, which could partially block the pores.

9) The abbreviation GVL should be clarified somewhere in the text, not only in the Scheme 1.

The authors thank the suggestion of the Reviewer. The abbreviation GVL was introduced in Scheme 1.

10) Figure 8 + 9 – the legend clarification is missing – “ND”, “u”

Following the suggestion of the Reviewer, the term ND was introduced in Figure 7 as “non-detected products”, while the legend -u was introduced in the manuscript on page 14 as “These spent catalysts were labelled with the final letter -u.”

11) The information, how the reused experiment was done is missing. What was the procedure between each step? Was the catalyst washed/dried? Were the losses of catalysts amount considered? This can be connected with the product found only in the second run but not in the third run.

According to the advice of the Reviewer, the methodology for the reusability test of the catalysts was included in the catalytic test section (4.4.):

In the case of the reusability studies, once the reaction was finished, the reactors were cooled down, extracting an aliquot of the reaction solution by decantation to be analyzed by GC. Then, the catalyst was filtered and washed with the solvent used in the reaction (2-propanol). Finally, the catalyst was dried before performing the next catalytic cycle.”.

12) Which data supports the mention, that Nb2O5 is well dispersed in Nb-SiO2?

The catalyst based on NbOx supported on a commercial silica was prepared by using similar methodology that was used for mesoporous silica. The aim of the synthesis of Nb-SiO2 catalyst was to highlight that the Nb-based catalysts supported on mesoporous silica display a better catalytic behavior. Considering these premises, the objective of the present study was the study of several NbOx-based catalysts with different textural properties. In addition, we previously prepared this type of catalyst (Nb-SiO2) in other works in which the dispersion of Nb2O5 species was suitable under the same synthetic procedure (García-Sancho et al. Molecular Catalysis 2017, 436, 267–275).

Round 2

Reviewer 3 Report

Comments and Suggestions for Authors

I have no additional questions. All my pevious notes were sufficiently supplemented and explained.